# Investigating and Compensating for Periphery-Center Effect among Commercial Near Infrared Imaging Systems Using an Indocyanine Green Phantom

Johanna J. Joosten [1,2], Paul R. Bloemen [2,3], Richard M. van den Elzen [2,3], Jeffrey Dalli [4], Ronan A. Cahill [4], Mark I. van Berge Henegouwen [1,2], Roel Hompes [1,2,*] and Daniel M. de Bruin [2,3]

1 Department of Surgery, Amsterdam UMC Location University of Amsterdam, Meibergdreef 9, 1100 DD Amsterdam, The Netherlands
2 Cancer Center Amsterdam, Imaging and Biomarkers, 1100 DD Amsterdam, The Netherlands
3 Department of Biomedical Engineering, Amsterdam UMC Location University of Amsterdam, Meibergdreef 9, 1100 DD Amsterdam, The Netherlands
4 Department of Surgery, Mater Misericordiae University, Hospital, 47 Eccles Street, D07 R2WY Dublin, Ireland
* Correspondence: r.hompes@amsterdamumc.nl; Tel.: +31-(0)20-566-9111

**Abstract:** Near infrared imaging (NIR) camera systems have been clinically deployed to visualize intravenous injected indocyanine green (ICG) spreading through the vascular bed, thereby creating the ability to assess tissue perfusion. While standardization is key to make fluorescence angiography (FA) comparable and reproducible, optical characteristics like field illumination homogeneity are often not considered. Therefore the aim of this study is to investigate light distribution and the center-periphery effect among five different NIR imaging devices in an indocyanine green phantom. A $13 \times 13$ cm fluorescence phantom was created by diluting ICG in Intralipid (representing 0.1 mg/kg dose in an 80 kg reference male), to evaluate the overall spatial collection efficiency with fluorescent modalities of five different NIR camera systems using a 0-degree laparoscope. The fluorescence signal from the phantom was quantified at a fixed distance of 16 cm using tailor-made software in Python. The results showed considerable variability in regard to light distribution among the five camera systems, especially toward the periphery of the field of view. In conclusion, NIR signal distribution varies between different systems and within the same displayed image. The fluorescence intensity diminishes peripherally away from the center of the field of view. These optical phenomena need to be considered when clinically interpreting the signal and in the development of computational fluorescence quantification.

**Keywords:** near infrared imaging; phantom; image guided surgery; light distribution

## 1. Introduction

Easier access to intra-operative near infrared (NIR) imaging has resulted in widespread use of this technology, allowing surgeons to see beyond the visual spectrum [1]. As an emerging optical imaging technique, NIR imaging has already shown clinical benefits in various surgical practices. NIR camera systems have been deployed to visualize intravenous injected indocyanine green (ICG) spreading through the vascular bed, thereby creating the ability to assess tissue perfusion, demarcate tumor tissue, and visualize vital structures [2,3]. When bound to blood plasma, ICG has a peak spectral absorption of around 800 nm and emits fluorescence at longer wavelengths [4]. In gastrointestinal surgery, indocyanine green fluorescence angiography (ICG-FA) has mainly been used to tackle perfusion-related complications, such as anastomotic leakage [5,6]. However, several studies report variable results in reducing the anastomotic leakage rates by using ICG-FA [5–8]. This may be due to the inter-user variation in interpretation of the FA signal, which remains mainly subjective using visual assessment only and is therefore associated

with a considerable learning curve [9,10]. However, while clinical use of fluorescence imaging systems is rapidly growing, the appropriate knowledge for reliable image interpretation is lagging. Essential in overcoming these challenges for broad effective implementation of ICG-FA, is standardization of fluorescence assessments to make them more comparable and reproducible. Standardization and protocolization enable the possibility of quantifying the fluorescence signal, which is an important focus of research in this field [11].

When standardizing the fluorescence measurement, tracer administration (dosage, volume, infusion rate), working distance, and ambient light are often taken into consideration [12]. To accurately interpret and quantify the fluorescence signal being displayed, a thorough understanding of the underlying physics is necessary. Factors such as distance, movement, and the relationship between the center and periphery can all have a significant effect on the intensity of the signal. These factors have received little attention by clinicians while it directly influences the fluorescence signal and may hamper correct interpretation. In clinical practice, for instance, the bowel located in the center may show more fluorescent intensity than a more proximal segment located in the periphery of the field of view (Figure 1) [13].

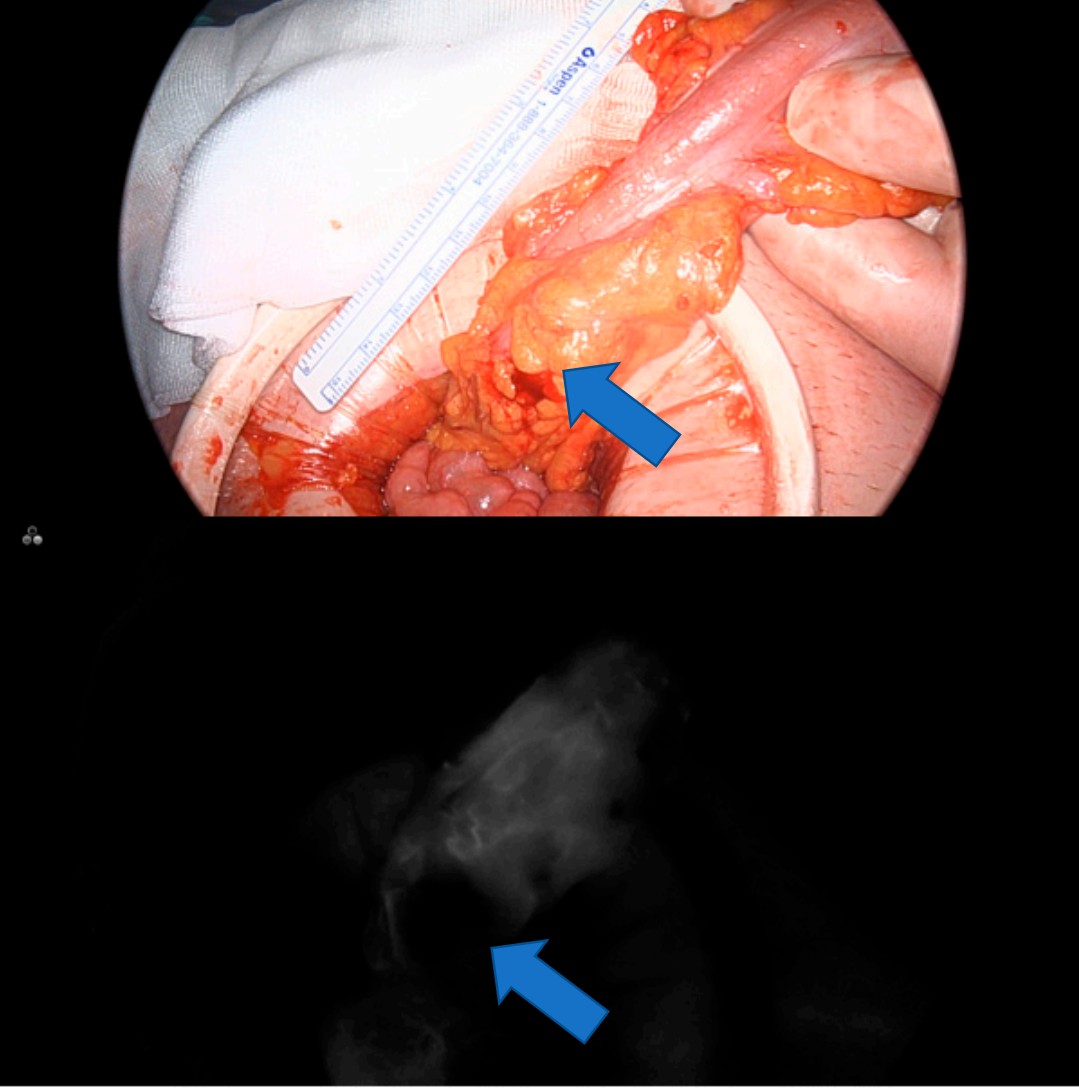

**Figure 1.** A segment of colon during a transanal total mesorectal excision. The top image is the white light image and the image below, the corresponding fluorescent image, demonstrating less fluorescent enhancement at a more proximal bowel segment (see blue arrow).

For other medical imaging modalities, universal standards are described to benchmark their performance. In order to do so for NIR imaging, solid tissue-mimicking phantoms have been developed to characterize NIR imaging systems [14,15], however, these papers have not yet been translated into applicable solutions for interpretation by clinicians. Moreover, so far, parameters such as the illumination homogeneity, the resolution, or the dependency of fluorescence intensity on tissue optical properties are not generally comprehensively addressed in the phantoms built. The studies that do address inhomogeneities of light distribution plot the intensity measurements only for a few locations (i.e., five reflective wells placed in the corners and center of the phantom), therefore, it is not possible to perform a correction of the recorded pixel data [16]. To better understand this periphery-center effect, which is observed in clinical practice, and promote awareness among clinicians and the potential consequences on fluorescent parameters, we investigated the light distribution within the field of view of five different commercial NIR imaging devices using a fluorescent phantom. We aimed to correct the data for illumination field distribution and light collection aberration known as vignetting to facilitate objective quantitative comparison of fluorescence in a flat field between systems.

## 2. Methods

### 2.1. Phantom Preparation

The phantom was created in a container of 13 × 13 cm, with a height of 6 cm. In total, 0.5 L of fluorescent phantom was created, resulting in a homogenous dilution within the container of 13 × 13 × 3.8 cm. The phantom was manufactured by heating 230 mL MilliQ ultrapure water with 20 g agarose (Sigma Aldrich, St. Louis, MO, USA, A9539) to generate a 4% agarose dilution. After the dilution was cooled to 60 °C, 250 mL of 20% stock Intralipid (Fesenius Kabi, Bad Homburg, Germany) was added to create a 10% Intralipid concentration, mimicking light scattering by tissue. A total of 285 μg ICG (Verdye, Diagnostic Green, Aschheim, Germany) was added to the mixture, and stirred for 10 min at 60 °C. The ICG concentration was calculated based on an individual of 80 kg, with 7 L blood [17]. The administered concentration of ICG we use in our clinical protocol is 0.1 mg/kg patient weight. With 7 L blood, the administered amount would correspond to 8 mg; for our phantom, this would result in 570 μg ICG. Several phantoms were prepared with different ICG concentrations, and the phantom with a concentration of 285 μg ICG was used as we obtained a perfectly homogeneous distribution.

### 2.2. NIR System Assessment

Five different clinical NIR camera systems (see Supplementary Table S1) were assessed using the set-up shown in Figure 2. These systems were: (1) Intuitive Surgical Inc. (Sunnyvale, CA, USA), XI Firefly, (2) Olympus (Tokyo, Japan), Visera Elite II, (3) Stryker (Kalamazoo, MI, USA) (Novadaq), AIM laparoscope, (4) Quest Medical Imaging (Wieringerwerf, The Netherlands), Quest Spectrum, (5) Stryker, Stryker 1688.

### 2.3. Fluorescent Assessment with the Phantom

IRB approval or written consent was not necessary as these assessments did not involve patients or patient characteristics. In theater, under fluorescent angiography conditions (dimmed surrounding light) the phantom was placed on an operating table. The laparoscope was fixed onto a mechanical holding arm (MOFIXX laparoscope holder, Alphatron Surgical, Rotterdam, The Netherlands) at a distance of 16 cm from the phantom surface, ensuring a stable position throughout the experiment. The distance was measured with a laser distance measurement system (Leica Distro D2, Leica Geosystems, St. Gallen, Switzerland). In this manner, the camera was fixated in the middle of the field of view (FoV). The recorded videos were transferred to a laptop for analysis.

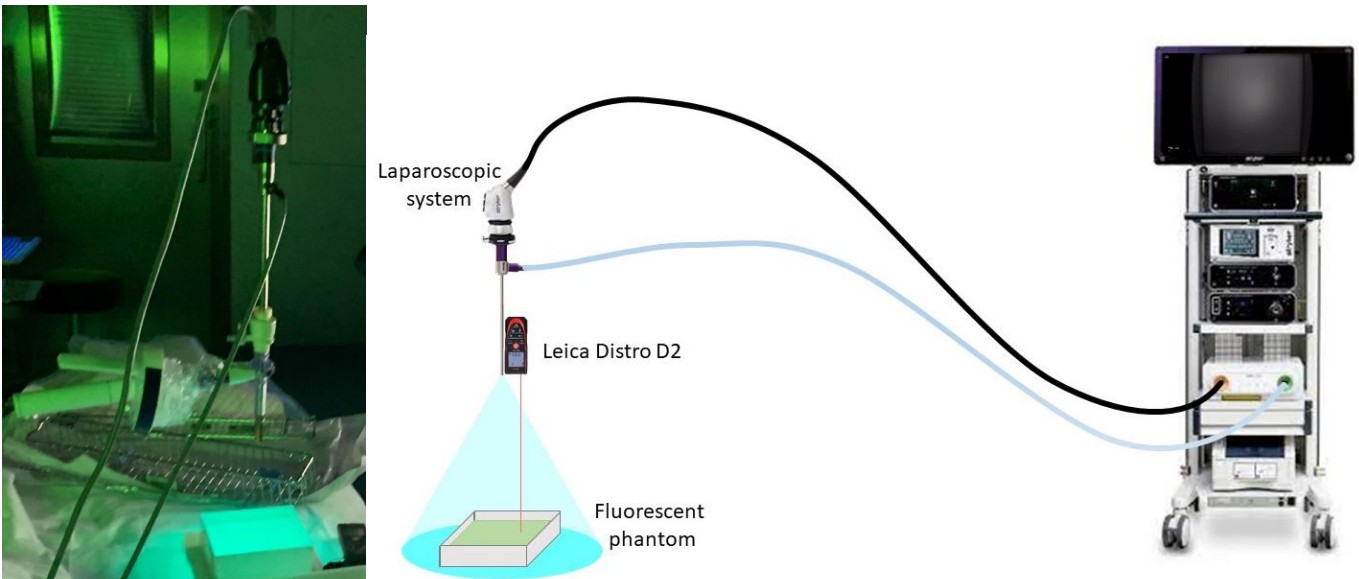

**Figure 2.** Experimental set-ups. The image at the left represents the phantom placed at 0-degrees with the endoscopic camera fixated at 16 cm.

### 2.4. Signal Quantification

Regions of interest (ROIs) in the video recordings of the phantom were analyzed by tailor-made software using Python v3.8 programming language (Python Software Foundation, https://www.python.org/, accessed on 10 September 2022). After calibrating the surface area in the video image using the known measurements of the phantom (see Supplementary Figure S1), a grid of 23 × 23 segments with a segment size of 5 mm was projected on the data within the phantom. While running the software the mean fluorescence intensity was calculated within each cell. The percentage of fluorescence signal loss (FSL) in the FoV was calculated by

$$\text{FSL} = 1 - \frac{F_{\text{low}}}{F_{\text{high}}} \times 100\%$$

in which $F_{\text{low}}$ denotes the lowest fluorescent intensity in an ROI and $F_{\text{high}}$ the highest.

For an objective quantitative comparison of fluorescence between systems, it is important to correct the data for influences, such as the point spread function of the used optics, illumination differences, and camera sensitivity of each individual system. This can be achieved using a flat-field correction known in microscopy. To that end, data from the phantom measurements can correct each system using a FIJI (open source imaging process software, version 2.0.0-rc-56/1.51 h, https://imagej.net/, accessed on 15 October 2022) pixel-by-pixel based implemented flat-field correction function as described by Schindelin et al. [18,19].

$$\text{i2 new} = \frac{\text{i1}}{\text{i2}} \times \text{k1} + \text{k2}$$

In which i1 is the original image, i2 is the phantom image, k1 is the mean intensity of the phantom image and k2 the mean intensity of a dark image (in our case left to 0). We used as an example a phantom measurement of the light distribution measured by a Stryker 1688 imaging device and a fluorescence angiogram acquired during a colon resection by the Stryker 1688.

## 3. Results

### 3.1. Light Distribution

A large variability in light distribution was seen within each camera system; the highest intensity was observed in the middle, with FSL toward the periphery of the image. In Figure 3 the light distribution was shown with the orientation of the phantom of the *x*-axis and normalized fluorescence intensity on the *y*-axis in order to portray the signal loss. This signal loss toward the periphery of the FoV was especially observed with the Firefly system (up to 60% loss at the periphery of the FoV) and the Stryker 1688. When the phantom was placed in the middle, the light intensity distribution was skewed to the right with 40% loss to the left edges. Figure 4 shows a schematic 3D representation of the light distribution. Conversely, the Quest system shows a higher intensity in the periphery compared to the center. The camera systems with the most homogeneous light distribution were Pinpoint and Olympus. The percentage FSL per system is shown in Table 1.

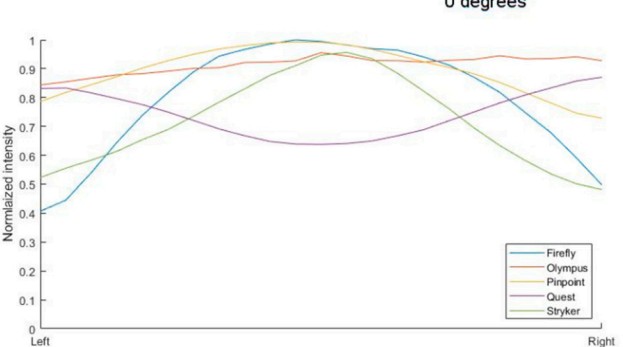
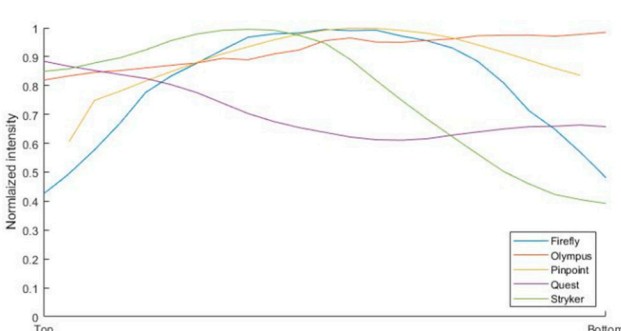

**Figure 3.** Four longitudinal cross-sections of the light distribution. On the *y*-axes, normalized intensity (maximum fluorescent intensity set at 100% and background noise at 0) is shown and on the *x*-axes the orientation of the phantom, which takes 13 cm from left to right. The top two images show the light distribution with the phantom placed at a zero degree.

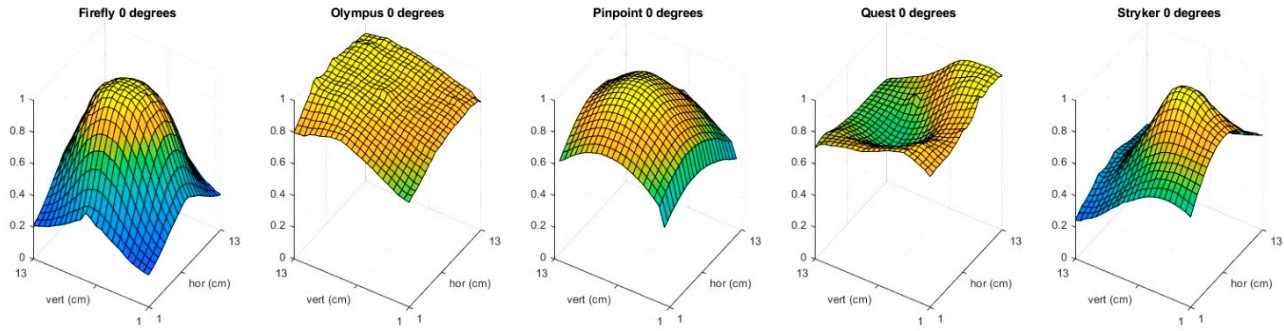

**Figure 4.** 3D representation of the light distribution among the five different cameras.

**Table 1.** Percentage of fluorescence signal loss per imaging device.

| Imaging Device | Highest Intensity (AU) | Lowest Intensity (AU) | FSL% |
|---|---|---|---|
| Firefly | 114 | 23 | 80 |
| Olympus | 86 | 57 | 34 |
| Pinpoint | 174 | 76 | 57 |
| Quest | 175 | 107 | 39 |
| Stryker | 107 | 26 | 76 |

### 3.2. Correction

Correction of the original data for influences such as the point spread function of the optics, illumination differences, and camera sensitivity was achieved by employing

a flat-field correction executed in FIJI. Figure 5 shows the acquired phantom data of the Stryker 1677 camera on a clinically obtained fluorescent image of a segment of colon during a transanal total mesorectal excision at the top left, with a red line indicating the location of the intensity distribution shown in the adjacent graph. Both adjacent graphs depict the intensity distribution and Locations 1 and 2 before and after correction.

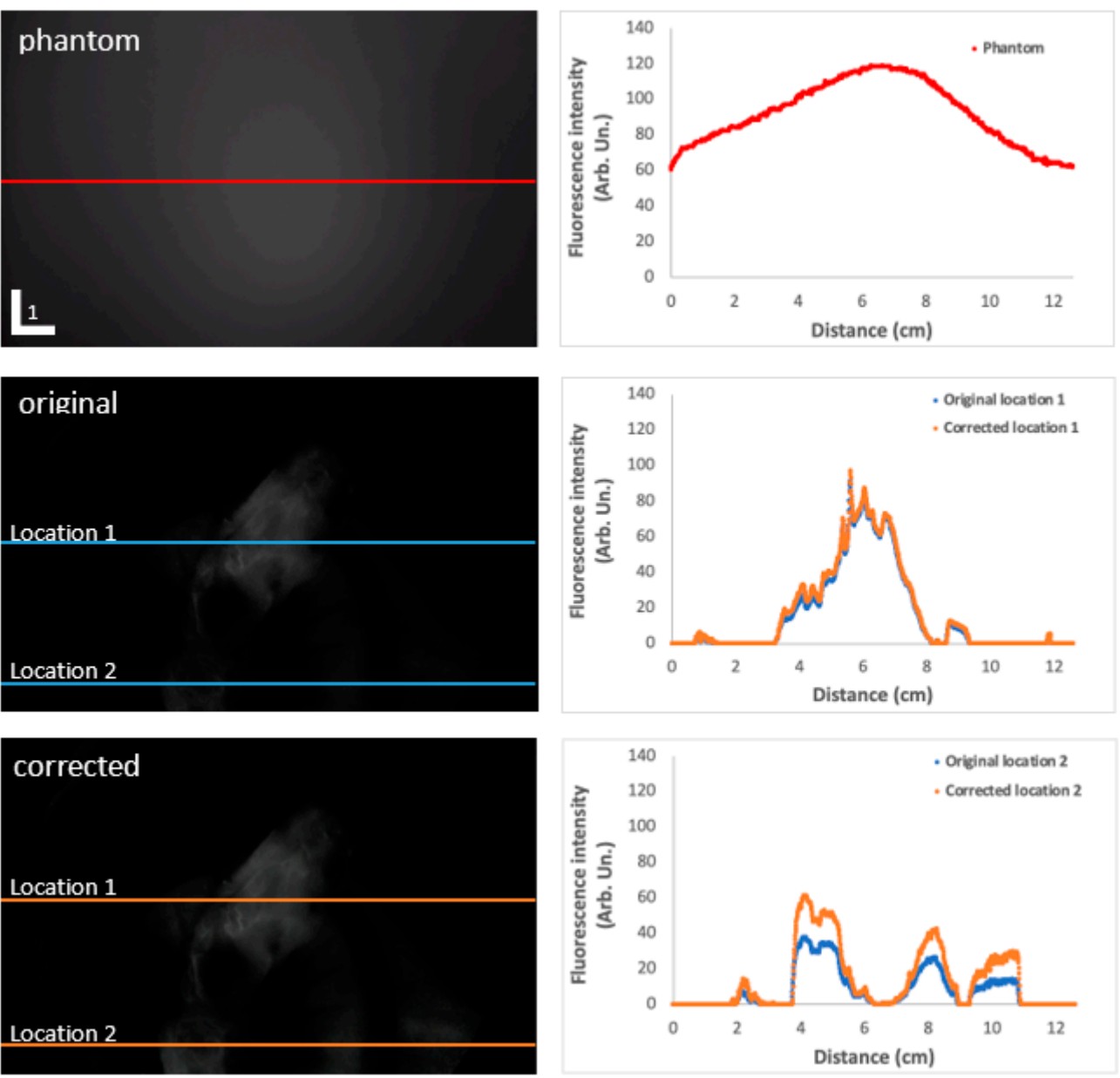

**Figure 5.** Top left image showing the phantom using the Stryker 1688 camera. The red line indicates the position of the adjacent graph with a fluorescent intensity distribution. The middle and lowest images both show the fluorescent image of a segment of colon during a transanal total mesorectal excision before and after correction. The blue and orange lines indicate the location on the intensity distributions plotted in the two adjacent graphs for the central and peripheral part of the image both before (blue) and after (orange) correction. The corrected result is therefore a more realistic representation of the fluorescence distribution in the tissue.

## 4. Discussion

The goals of improving fluorescent assessments and their interpretation are similar to those in science in general; achieving objectivity, reproducibility, and comparability. In this

study, a fluorescence imaging standard was used that incorporated ICG to test the center-periphery effect of five commercially available imaging systems indicated for use with ICG. The results of this study demonstrate a considerable variation in light distribution within the FoV; with decreasing fluorescence intensity toward the periphery down to 60% in relation to the highest intensity. Among the five different commercially available NIR camera systems, considerable variability exists regarding fluorescence intensity and sensitivity. These features should be considered and if possible corrected for while interpreting the fluorescent signal clinically and selecting ROIs for quantification.

Recent reviews have stated that inflow parameters have superior clinical performance over intensity parameters, due to immunity to changes in camera distance and angulation [12,20]. Our findings emphasize these results, as dynamic flow parameters are less likely to be affected by the RoS selection and thus inhomogeneous illumination fields. Despite the fact that for quantification the camera is kept stable in terms of distance, movement due to breathing or peristalsis is problematic and challenging to eliminate. Furthermore, by utilizing normalization and correction for the light distribution, one could modify the slope parameter to be based on relative changes in fluorescence intensity, significantly reducing the impact of several of the above-mentioned challenges that affect the measurement and differences in camera systems.

This study addresses the periphery-center effect in the flat field only and does not take different distances or the morphology of the tissue into account. In this study, it was decided to standardize the distance and optical properties to remove the confounding effects which lightscattering and absorption can have on fluorescence. These factors require other, more sophisticated solutions. The aim of this study was to quantify the impact of the periphery-center effect in the flat field and to correct it as a first step.

Recent work has demonstrated that significant variations in performance between NIR cameras exist, with each camera system being used at different distances and speeds of movement [21]. The results indicated in this study were focused on defining the ideal optical distance (FoV) and distance per camera demonstrating that each device has its own ideal set-up.

Our results demonstrate that there are significant variations among different camera systems. It is likely that other camera systems not included in our testing would also exhibit these differences. This makes it challenging to establish consistent fluorescent parameter thresholds across different camera systems. It is important to note that most commercial systems are designed primarily to provide surgeons with information about perfusion, rather than to quantify the signal or for inter-system compatibility. Currently the control over system parameters such as gain, camera integration time, and illumination intensity is limited. However, recent developments such as manual gain settings, protocols for fluorescent assessments that include tracer dosage, and standardization of distance aim to quantify and calibrate the results to some extent in the future. As shown in this study, the periphery-center effect impacts quantification outcomes if one is not aware of this while choosing ROIs (i.e., the fluorescent intensity may decrease significantly in the periphery, also represented by the FSL%). In clinical use, even though the camera is likely to be focused on the ROI at its center, one should take this into account and possibly correct for it. As the number of commercially available optical imaging systems has greatly increased, this technology is now being used by more inexperienced surgeons who occasionally lack knowledge of the fundamental principles and pitfalls of optical imaging, making it difficult to interpret imaging data in a way that is accurate, precise, reproducible, and reliable. Raising awareness that using and comprehending optical imaging modalities requires training and is, therefore, a learning curve, could lead to the organic beginning of this new class of trained surgeons; the interventional imaging surgeon. This is not novel because it is comparable to traditional imaging techniques, which today each have their own class of medical professionals, namely the nuclear physician and the radiologist.

However, being able to compare fluorescent thresholds between patients and/or systems is a future goal as well as a challenge. The flat field corrected illumination differences

and optical behavior of the NIR imaging devices in these studies can be incorporated together with patient data in (deep learning) algorithms to automate quantitative/artificial intelligent ICG perfusion angiogram classifications to facilitate its interpretation for all clinicians, not only the prior mentioned interventional imaging experts. These algorithms incorporated in robotic or laparoscopic consoles might propose resection lines after NIR assessment on the live tissue based on patient-, optical-, fluorescent characteristics.

This study is limited first for not truly mimicking the clinical situation, as the phantom is entirely flat. However, correcting for different angulation, depth, and tissue morphologies is hard to summarize in one formula. Second, it was conducted using an ICG phantom, however, both the methodology as the message of this study apply to other fluorescent tracers of the same emission peak. In addition, during this study we chose to perform all the assessments with one fixed distance so as to have the entire phantom in the FoV. To achieve this, a distance of 16 cm was set, however, during laparoscopy, especially in patients with a low BMI, a smaller distance is conventional and usually in the range of 10 cm. That said, the scope of this paper was not to mimic clinical use but to demonstrate the light distribution and impact of angulation. The experiments were executed in a completely dark environment as is the case in an abdominal cavity, and we assume that this has no influence on the correction, given that the image would have only 'zero' intensity pixels. Ideally, a fluorescent phantom offers comprehensive information on multiple systems' parameters with only one snapshot, but this phantom only addresses illumination and camera sensitivity. This study is also limited in assessing only the commercial camera systems we had at our disposal, whereas clinical practice is dependent on the fluorescence systems hospitals have available.

In conclusion, there was a considerable center-periphery effect within an FoV with sometimes not even peak intensity in the middle of the FoV, and this differs among systems. Other phantoms with only a few wells do not capture these illumination differences entirely, these features should be considered and if possible corrected for while interpreting the fluorescent signal clinically and before selecting ROIs for quantification.

**Supplementary Materials:** The following supporting information can be downloaded at: https://www.mdpi.com/article/10.3390/app13042042/s1, Table S1. Table demonstrating features of camera systems tested in this experiment. Data from user manuals, communication with company representatives or previously published assessments denote that data were not available or obtainable through previously mentioned methods. Figure S1. The software used to calibrate and analyze the illumination field by grid. First, the phantom size was calibrated by a red line after indicating its length (130 mm), then a grid of $23 \times 23$ size cells was placed inside the phantom.

**Author Contributions:** J.J.J. carried out the study and collected the results as well as being involved in study conception, design, and manuscript drafting. P.R.B., R.M.v.d.E. and J.D. were involved in study conception, design, and collection of the results. M.I.v.B.H., R.A.C. and R.H. were involved in the study conception and design. D.M.d.B. was senior author on the project contributing to study conception, design, result analysis, and writing. All authors have read and agreed to the published version of the manuscript.

**Funding:** This research received no external funding.

**Institutional Review Board Statement:** Not applicable.

**Informed Consent Statement:** Not applicable.

**Conflicts of Interest:** Johanna J. Joosten, Paul R. Bloemen, Richard M. van den Elzen, Daniel. M. de Bruin: have no conflicts or financial ties to disclose. Jeffrey Dalli is employed as researcher in the DTIF and is recipient of the TESS scholarship (Malta). Mark. I. van Berge Henegouwen: unrestricted grants from Stryker and consultancy for Johnson and Johnson, Als, Surgical, Mylan, Braun, and Medtronic. All fees and grants paid to the institution. Ronan A. Cahill; speaker fees from Stryker Corp, Olympus and Ethicon/J&J, research funding from Intuitive Corp and Medtronic and from the Irish Government (DTIF) in collaboration with IBM Research in Ireland, and from EU Horizon 2020 in collaboration with Palliare. Roel Hompes: unrestricted grant and materials from Stryker European Operations B.V.

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
