# Peer review of "Investigating and Compensating for Periphery-Center Effect among Commercial Near Infrared Imaging Systems Using an Indocyanine Green Phantom"

_applsci, doi:10.3390/app13042042_

Round 1

Reviewer 1 Report

This is a well-written comparison of fluorescence-capable surgical imaging systems that details what many have observed clinically in addition to providing an algorithm for correcting aberrations.   While it would be interesting to compare the raw imaging data (pre-processed images) between systems, manufacturers do not allow user access for the most part.    

Your article makes a nice contribution to the field, and points out that for the most populous imaging systems out in the world, it pays off to keep your region of interest in the center of the endoscopic image.

I have included suggested edits within the attached PDF.

Reviewer 2 Report

In this manuscript the authors aim to analyze and compensate the effects of camera characteristics and imaging angle in clinical fluorescence imaging of ICG in angiography. They employ five different commercial clinical systems and a custom-made fluorescent phantom in order to obtain the images. Comments on the manuscript appear below:

 -        The title of the manuscript is much more generic than the scope of the intended analysis. The manuscript is completely focused on ICG-FA, and this would be clearly stated in the title, as it is not clear if the results could be extended to other fluorophores.

-        The introduction of the manuscript mentions ICG-FA and the problems regarding standardization of the images obtained. In general, it is quite short and lacks much more detail in my opinion. For instance, the so-called physical factors are not supported by references in which these parameters might have been considered before. There is even an example of an image in this part that seems to be out of the general context, and the information it shows is not clear with the blue arrow. A deeper detailed analysis of the problem, of the previous approaches that could have dealt with it before, and the specific contribution of this work regarding the previous works needs to be included.

-        The configuration of the experimental setup is quite undefined for the aim of the manuscript. Phantom preparation includes enough detail, but for instance the final selection of the amount of ICG as half of the calculated one is not clear. The experiments are carried out by clinical commercial devices. This fact makes the analysis quite difficult, as there is no absolute control on the possible dynamic characteristics and potential post-processing of the images obtained by each of the systems. There is even no description of each of the systems in terms of technical data that could influence the results. Figure 2 shows just two images of one of the systems in the operating room, with the photography corresponding to 30º angle being specially not clear. There is no diagram of the setups employed, as it would be necessary to completely understand the experimental system. There is also no correction of the distance to the sample as a function of the system and its particular optical characteristics, as this distance was decided to be fixed. As a consequence, the experimental approach is not appropriately described and taken into consideration in the measurements.

-        The metrics employed for the quantification of the results are also not clear. It is mentioned that the surface area in the video image is calibrated with the known measurements of the phantom, but it is not clear what this means and must be explicitly shown. The percentage of signal los seems to be quite dependent on the size of the grid, and there is no reference to this point. There is a reference to point spread function, illumination and sensitivity of the cameras, but no data on these aspects is provided nor analyzed. A flat-field correction is mentioned, with a definition that is not clear. Are the operations in the equation being made on a pixel by pixel basis on the images? Why is the dark image set to 0, when in any imaging technique it is usually very important to balance the contribution of pixels to a dark image?

-        Regarding the results, the variability of light distribution, that conforms Figures 3 and 4, seems to be obtained directly from the images of the phantom, is that correct? If this is the case, could they be affected by photobleaching? Which irradiances are used in each of the systems? For how long are the images obtained? Are the images obtained on the same phantom with all the techniques? If so, these effects could be affecting the results. The distribution of light directly on the images is quantified exclusively by the previously defined fluorescence signal loss, and some specific percentages with no clear calculation methodology. It is not clear how and to what extent these results may influence the clinical praxis in the field. In general, there is a lack of adequate quantification of the differences. This is particularly evident in the case of angulation, where practically no quantitative results are shown.

-        The correction proposed based on the phantom for the images of the colon in Figure 5 is not clear enough. Which system where the images obtained from? What are the new quantification values for the corrected image when compared with the original one? What would be the potential impact in clinical practice?

-        As it was expected from the results obtained, the discussion and conclusions are quite open and generic, with no clear aim being exposed nor solved. In my opinion the reader could not find a clear contribution of the work as it is now formulated and conceived.

 As a consequence, and although the topic is of great interest, the present manuscript in my opinion does not present a contribution to the field. I would advice the authors to completely redesign the study with clearer objectives, a clear experimental analysis with appropriate parameters, and a much deeper technical analysis of the characteristics that could influence the clinical images.

Reviewer 3 Report

applsci-2064464

The manuscript characterizes images of fluorescence phantoms consisting of indocyanine green (ICG) and intralipid mixtures using various NIR camera systems. The manuscript argues that in order to standardize the fluorescence measurement, it is necessary to go beyond body mass adjusted ICG dosage and detector-tissue distance. The manuscript demonstrates the importance of detector angle and introduces corrections to facilitate quantitative comparisons between systems.

This manuscript only shows that NIR systems have variations in illumination and detection, which was already known to exist. The “corrected” image (Figure 5 bottom) shows improved contrast but is not being assessed by any objective references or measures. Overall, the manuscript does not present or propose protocols for standardization.

Technical comments:

1)  Line 70: Please list optical properties of ICG.

2)  Line 83: The names of the NIR camera systems should be presented in a consistent format, e.g., model, manufacturer, manufacturer location. Relevant model numbers should be provided, e.g., of the camera and probe. Relevant technical parameters need to be included, e.g., excitation wavelength and source, camera QE and spectral range, spatial and spectral resolution, etc. At what wavelength are the fluorescence images collected?

As is, there is not sufficient information to reproduce the results.

3)  Line 90: Pictures are too small for understanding the set up. On the picture, clearly indicate the location of the phantom. Clearly indicate the probe-to-phantom separation and the angular orientation. The 30-degree angle is not clear.

4)  Line 122: I could not easily find this equation in reference 12. Please clarify how the reference is being used and how the correction is obtained. Also, is the correction applied on a pixel-by-pixel basis? Which phantom image is used to correct which sample image?

5)  Line 138: What is the difference between the left and right columns of Figure 3? Are they two representative cross sections? Can quantitative measures of reproducibility be provided?

How are the curves being normalized?

6)  Line 146: What is the usefulness of FSL% parameter? Is it only used as a measure of flat-field-ness?

7)  Line 166: “The corrected result is therefore a more realistic representation of the fluorescence distribution in the tissue.” What is the justification for saying it is more realistic? True, weaker features are more visible after the correction, but how is it known that it is more objectively correct? 

Additional comments

1) Refs 1, 5, 8, and 13 are incomplete.

2) Fluorescence images (Figures 1 and 5) are difficult to see.

Round 2

Reviewer 2 Report

Although the introduction has been improved, the contribution of the authors in the manuscript is still not clear. It is said that “…(previous) papers have not yet been translated into solutions for commercial available clinical systems”. This seems to be a lab to market issue, not a scientific-technical one. The other parameters mentioned, such as illumination homogeneity, resolution or dependence on optical properties, are not specifically and clearly exposed in the existing solutions compared to the present one. Consequently, the novelty of the manuscript is still not clearly assessed in my opinion, and this is a fundamental point.

Regarding methods, the selection of the phantom (with an amount of 285 micrograms of ICG) is motivated just by the fact that it was the only one with a homogeneous distribution that the authors could manage. There are no variations of the angle of the system with respect to the phantom/tissue, or on the distance. The issues about the potential post-processing of the commercial systems are also not solved. The proposed correction is just based on the illumination inhomogeneity, measured on the phantom, with no considerations on several angles, distances, or tissue morphologies that could locate points at different positions (for instance, in the case the tissue is tubular).

The results are not adequately quantified, in terms of accuracy or improvement regarding the standard approach. The figures are not discussed deep enough, leaving almost all the explanations to the figure captions.

The topic of the manuscript is of great clinical interest. However, in my opinion, the topic is treated in a quite superficial way, and the revised version of the manuscript has not accomplished most of the comments of the initial review. This study should be made in a more systematic, deep and complete way in order to be of real interest to the community in my opinion.

Author Response

  1. Although the introduction has been improved, the contribution of the authors in the manuscript is still not clear. It is said that “…(previous) papers have not yet been translated into solutions for commercial available clinical systems”. This seems to be a lab to market issue, not a scientific-technical one.
    1. We appreciate your bringing this to our attention. We concur that this is a crucial issue for companies in the near-infrared imaging industry to address in order to apply these technologies in clinical settings. Currently, many systems lack the ability to adjust the gain. Our paper addresses the significant differences in light distribution and peripheral-center effects between commercial systems, which impede clinical studies and use. Addressing these limitations may lead to the inclusion of manual gain settings, allowing for better calibration between systems. We believe our study highlights the need for transparency in regards to these optical properties.

  1. The other parameters mentioned, such as illumination homogeneity, resolution or dependence on optical properties, are not specifically and clearly exposed in the existing solutions compared to the present one. Consequently, the novelty of the manuscript is still not clearly assessed in my opinion, and this is a fundamental point.

  1. This study only addresses the periphery- center effect in the flat field only and does not take different distances or the morphology of the tissue into account. In this study, it was chosen to standardize the distance, and optical properties to remove confounding effects which light scattering and absorption can have on fluorescence. These factors require other, more sophisticated solutions. The aim of this study was to quantify the impact of the periphery- center effect in the flat field and to correct it as a first step. This statement has been added to the discussion. We have chosen to remove the angulation bit from the manuscript and only highlight the periphery-center effect.

  1. Regarding methods, the selection of the phantom (with an amount of 285 micrograms of ICG) is motivated just by the fact that it was the only one with a homogeneous distribution that the authors could manage.

  1. Correct, if we would perform the study again we would perform the assessments with different concentrations. However unfortunately this was the only concentration we could work with due to inhomogeneity of our other phantoms.

  1. There are no variations of the angle of the system with respect to the phantom/tissue, or on the distance.

  1. The reviewer is correct by stating that there is no variation in the angle between phantom and system. However, in practice the angle between the tissue and the laparoscope is unknown and therefore is difficult to correct. We only show the effect of illumination inhomogeneity and vignetting as a combined system parameter which can be corrected for as a first step. In order to emphasize the extend of this manuscript, we specified this more clearly in the introduction and limitations part of the discussion.

  1. The issues about the potential post-processing of the commercial systems are also not solved.
    1. As stated in the discussion, due to the commercial nature of the systems it is impossible to have control over most of the post-processing parameters such as gain, camera integration time and illumination intensity. For this reason we used a standardized setting to be as objective as possible.

  1. The proposed correction is just based on the illumination inhomogeneity, measured on the phantom, with no considerations on several angles, distances, or tissue morphologies that could locate points at different positions (for instance, in the case the tissue is tubular).

  1. The reviewer is correct in the statement that only the correction is based on illumination inhomogeneity. However based on how we performed the measurements, vignetting is also included in the overall light distribution of the complete system. We do not claim to solve the influence of tissue geometry i.e. tubular tissue and the effect this has on the optical performance of the system, due to unknown spatial geometry of the sampled tissue. In order to emphasize the extend of this manuscript, we specified this more clearly in the introduction and limitations part of the discussion.

  1. The results are not adequately quantified, in terms of accuracy or improvement regarding the standard approach. The figures are not discussed deep enough, leaving almost all the explanations to the figure captions.
    1. We are not sure what the reviewer means with not adequately quantified. We described the quantification software we used to quantify the signal on the phantom and used the FSL% in order to quantify the periphery-center effect. We addressed the figures more thorougly in the manuscript.

  1. The topic of the manuscript is of great clinical interest. However, in my opinion, the topic is treated in a quite superficial way, and the revised version of the manuscript has not accomplished most of the comments of the initial review. This study should be made in a more systematic, deep and complete way in order to be of real interest to the community in my opinion.
    1. We agree with the reviewer that the manuscript is of great clinical interest. The reviewer states that this study should include effects as angulation and tissue geometry etc. however this is beyond the scope of the current study. This is two-fold, namely, a) quantifying the periphery center effect in the flat field and b) the possibility to correct for this effect, which is the first step.

Reviewer 3 Report

Supplementary Table 1 should be cited in the "NIR system assessment" section of the manuscript. 

Also, the company and system name in the text do not all match the company and system name in the table.  This is confusing.  For example,

Line 115: "Olympus, Visera Elite II system, Ireland" matches the table.  

Line 115: "NIR system, Pinpoint, Stryker, USA laparoscopic system" does not match the table.

Thirdly, the format in the text aren't all parallel.  The first is in the format "[company], [system], [country]" and the second is in the format "[unknown], [camera], [company], [country], [lap]."

Minor perhaps, but it is confusing.  This was mentioned in my previous review and not fully addressed.

I have no new comments.

Author Response

Thank you for addressing this, and we agree this is inconsistent and confusing. We have changed and clarified this in the manuscript.